# Few-Shot Learning for Misinformation Detection Based on Contrastive Models

**Peng Zheng [1,†], Hao Chen [1,†], Shu Hu [2] , Bin Zhu [3], Jinrong Hu [1,*], Ching-Sheng Lin [4], Xi Wu [1], Siwei Lyu [5] , Guo Huang [6] and Xin Wang [7,*]**

1 School of Computer Science, Chengdu University of Information Technology, Chengdu 610225, China
2 Department of Computer Information Technology, Purdue University in Indianapolis, West Lafayette, IN 47906, USA
3 Microsoft Research Asia, Beijing 100080, China
4 Master Program of Digital Innovation, Tunghai University, Taichung 40704, Taiwan
5 Department of Computer Science and Engineering, University at Buffalo, Buffalo, NY 14260, USA
6 School of Electronic Information and Artificial Intelligence, Leshan Normal University, Leshan 614000, China
7 Department of Epidemiology and Biostatistics, School of Public Health, University at Albany, Albany, NY 14260, USA
* Correspondence: hjr@cuit.edu.cn (J.H.); xwang56@albany.edu (X.W.)
† These authors contributed equally to this work.

**Abstract:** With the development of social media, the amount of fake news has risen significantly and had a great impact on both individuals and society. The restrictions imposed by censors make the objective reporting of news difficult. Most studies use supervised methods, relying on a large amount of labeled data for fake news detection, which hinders the effectiveness of the detection. Meanwhile, the focus of these studies is on the detection of fake news in a single modality, either text or images, but actual fake news is more often in the form of text–image pairs. In this paper, we introduce a self-supervised model grounded in contrastive learning. This model facilitates simultaneous feature extraction for both text and images by employing dot product graphic matching. Through contrastive learning, it augments the extraction capability of image features, leading to a robust visual feature extraction ability with reduced training data requirements. The model's effectiveness was assessed against the baseline using the COSMOS fake news dataset. The experiments reveal that, when detecting fake news with mismatched text–image pairs, only approximately 3% of the data are used for training. The model achieves an accuracy of 80%, equivalent to 95% of the original model's performance using full-size data for training. Notably, replacing the text encoding layer enhances experimental stability, providing a substantial advantage over the original model, specifically on the COSMOS dataset.

**Keywords:** contrastive learning; misinformation detection; deep learning; few-shot learning





## 1. Introduction

The rapid dissemination of news articles through social media platforms offers immediate access to information, but it also facilitates the rampant spread of misinformation driven by deceptive practices [1]. Misinformation poses significant challenges to both social media [2] and real society [3], and it has been a prevalent issue in recent years. A glaring example of this is the impact of misinformation during the COVID-19 pandemic, which created fertile ground for conspiracy theories. One particularly prominent conspiracy theory suggested that COVID-19 vaccines are poison and that the messenger RNA technology has not been tested yet and is harmful [4]. This rumor has long been debunked and is considered a baseless conspiracy theory. However, it has led people to resist the COVID-19 vaccine and waste public resources. Moreover, the widespread dissemination of false information has detrimental consequences, resulting in heightened anxiety across

various age demographics [5,6]. Therefore, it is crucial to curb the spread of fake news on social media, and the key lies in automated methods for detecting misinformation.

Detecting fake news and misinformation is a critical challenge in today's proliferation of social media, and there is a growing need for accurate and efficient approaches to address this issue [7]. Manual fact-checking is time-consuming and labor-intensive [8]. For this reason, various automatic approaches have been developed to combat fake news, such as those utilizing machine learning techniques like supervised classification models [8–13]. However, these models are mainly required to take advantage of high-quality labeled data, which may be scarce and not cover the full diversity of fake news content. In contrast, utilizing weakly supervised or unsupervised methods [14–19] does not require large amounts of labeled data and has significant advantages over supervised methods [20].

Many researchers have primarily focused on using Natural Language Processing (NLP) techniques to detect fake text-based content [21–24], often overlooking the fact that news articles frequently include both textual and visual elements. An illustrative example can be seen in Figure 1. In this figure, the genuine image caption is highlighted in the green box, while the false caption is highlighted in the red box. Such instances of misinformation, where images are paired with misleading captions, are easily generated or manipulated by AI-based tools and rapidly disseminated across the internet. Existing approaches, which extensively use textual semantic and syntactic similarities [25–27], have shown promising results, to some extent, in identifying fake news. However, these models tend to neglect the interactions between multiple data modalities, particularly the relationship between images and text, potentially hampering their accuracy. To address the limitations of these models, Aneja et al. [28] proposed a solution that combines both textual and visual data. They introduced the COSMOS dataset, which includes images along with two captions for each image. Their task involves predicting whether the two captions are both authentic to the image, essentially an out-of-context (OOC) classification problem. The baseline model in their work employed a convolutional neural network as an image encoder and a pre-trained language model as a textual caption encoder, achieving an 85% classification accuracy on the COSMOS dataset. However, it is worth noting that their model was trained on a large corpus, which can be less efficient and time-consuming. Additionally, the model's performance is significantly influenced by the choice of the pre-trained text embedding model (SBERT), as revealed by our investigation.

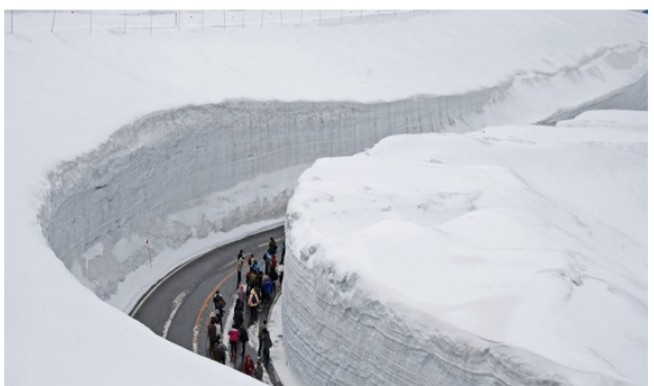

**(real) caption_1:** *Are These Towering Snow Walls in Massachusetts*

**(random) caption_2:** *The journey of Tateyama Kurobe in Japan*

**Figure 1.** An example of text–image pairing. For the image on the left, the text in the green box is the real(matched) caption, while the text in the red box is a fake (random) caption.

In this paper, we present an extension of our previous work [29] for out-of-context detection, applied to the COSMOS dataset. Our approach leverages a language–vision model based on contrastive learning [30]. It is a self-learning technique that demonstrates a remarkable ability to learn feature representations through the contrast of similar and dissimilar pairs of examples. These features are valuable for various downstream tasks, including image classification and language understanding. This approach has proven

particularly effective in scenarios where labeled data are limited or costly to obtain, as contrastive learning can exploit large volumes of unlabeled data for training. We used Aneja et al.'s [28] method as the baseline in our experimental evaluation. Compared to the original method, ours achieved 95% of the original model's performance with only 3% of the data. It possesses enhanced feature extraction capabilities, and the performance remains more stable after replacing the text encoding layer. This indicates that our proposed method outperforms the baseline on the out-of-context prediction. The main contributions of this paper can be summarized as follows:

- We have discovered that the baseline approach heavily relies on the pre-trained text embedding model, SBERT, while the image feature is ignored for the out-of-context (OOC) classification task. This reliance on textual features brings about the possibility of bias and distortion in the model's outcomes.
- We developed a novel model that integrates a contrastive learning component, which offers distinct advantages in capturing image feature representations. This proves particularly beneficial when confronted with limited training data.
- We conducted comprehensive experiments to evaluate our proposed approach, investigating different text encoding methods and examining the impact of varying training data volumes on model performance. We compared our method with baseline approaches. The results of these experiments indicate that our approach is more stable and superior to the baseline methods.
- Rather than an OOC task, we developed a classification model to identify whether a caption corresponds to the image, and it shows promising results.

It is important to note that most of the techniques we introduce are general and can be applied to various classification tasks. Specifically, our use of contrastive learning proves advantageous in scenarios with a scarcity of labeled training data. This paper is organized as follows: Section 2 reviews related work, and Section 3 describes our proposed method. Our experimental evaluation is presented in Section 4. The conclusion and future work are presented in Section 5.

## 2. Related Work

Online misinformation has gained significant attention in recent years, prompting extensive research efforts to combat this issue [31]. Bondielli et al. [32] classify information as either fake or rumors based on whether it has been verified by authoritative sources. Guo et al. [33] and Meel et al. [34] analyzed the distinctions among various terms related to misinformation on social media, such as hoax, disinformation, and fake news. Rather than delving into the intricacies of these different definitions, we aim to focus on machine learning techniques, particularly contrastive learning. In this context, we will provide a brief review of relevant research encompassing misinformation detection and contrastive learning.

### 2.1. Misinformation Detection

Misinformation detection is the information in the news that helps the model identify whether the statement is true or false. Early research mainly analyzed the text of unimodal data and primarily relied on classic supervised classification methods, such as support vector machine (SVM) [35–37], naïve Bayes [38], logistic regression [39], and decision tree [36,40]. Traditional feature representations such as bag of words and n-grams with TF-IDF are generally used, with semantic or syntactic information ignored due to individually treated features (word tokens). Motivated by the rise of word2vec and paragraph2vec, numerous recent studies [26,27,41–43] have delved into the use of distributed representations. In this approach, textual content is transformed into a compact vector using a language embedding model. Typically, this model is pre-trained on a broad language corpus, preserving essential language characteristics like syntax and semantics. Along with the continuous progress of technology, researchers have started to use deep learning techniques for feature extraction, which allows the model to learn features better than

previous feature extraction methods. For instance, Ma at al. [44] introduced a recursive neural network with a tree-like structure rather than a typically sequential propagation structure to learn features for fake news detection. To alleviate the problem of gradient vanishing, Chen et al. [45] developed a combination of a recurrent neural network and an attention mechanism. In addition, Shu et al. [46] proposed a framework named TriFN that takes account of social network information, such as the relationships between news publishers and news, as well as the interaction between users and news.

In addition to pure textual information, multimodal misinformation involving both images and text is widely spread online. In such cases, the image and text are not directly related, a phenomenon referred to as de-contextualization (also known as out-of-context pairing) [47]. Researchers extract features from different modalities and then fuse them to determine whether the information is fake or not. Singhal et al. [48] proposed the SpotFake model, which utilizes pre-trained VGG19 for extracting image features and BERT for extracting text features, harnessing the powerful performance of pre-trained language models. These two modal representations are then concatenated and used as input for the fake news discriminator. Singh et al. [49] extracted text features using RoBERTa and utilized EfficientNet to extract features from images after ELA analysis, aiming to examine the reliability of images in news articles. Qian et al. [50] proposed a model that leverages BERT to extract textual features and uses ResNet50 to extract image region features. It employs a dual-stream fusion approach to predict the authenticity of news articles. This paper, building upon previous research, places a strong emphasis on the use of contrastive learning to enhance the model's performance in situations with limited training data.

### *2.2. Contrastive Learning*

As deep learning techniques have progressed, the demand for large-scale data has gradually increased [51]. Taking CLIP [52] as an example, the amount of data required for its training has been as high as 400 million pairs of samples, while in comparison, the ImageNet [53] dataset, which is widely used in computer vision applications, contains only 1.3 million labeled data samples. However, contrast learning, as an unsupervised learning method, has excellent feature extraction capabilities without relying on a large amount of supervised labeled data. The core principle is that different data-enhanced representations of the same sample are clustered in the feature space, whilst the representations generated from different samples are pushed apart. This process can be viewed as creating an ordered structure in the feature space that brings similar samples closer together in the space and dissimilar samples farther away, thus helping machine learning models better understand the intrinsic structure and characteristics of the data. Contrast learning is a promising approach to learning feature representations when dealing with data imbalance and a lack of labeling information [54].

Contrastive learning has undergone continuous development, leading to the introduction of several highly effective models. Inspired by the results of supervised learning, Ye et al. [55] proposed the individual discrimination proxy task, which treats each image as a separate individual and aims to make the features of the same individual as similar as possible while ensuring that the features of different individuals are as dissimilar as possible. Impressive results have been achieved in unsupervised representation learning by leveraging this proxy task. Subsequently, He et al. [56] proposed using a queue to replace the memory bank in [55], eliminating the limitation of the dictionary containing negative samples imposed by the batch size. This replacement ensures consistency and has resulted in a significant improvement in model performance. Chen et al. [30] proposed SimCLR, which integrates previous techniques and introduces a creative addition of an extra encoding layer after obtaining the data. This additional layer is used to reproject the features before conducting contrastive learning. Caron et al. [57] proposed SwAV (Swapping Assignments between Views) by modifying SimCLR, which allows for data clustering, enabling the model to perform stable clustering directly on samples within a

batch. The model then utilizes the clustering results for contrastive learning, helping the model acquire meaningful sample representations.

## 3. Our Proposed Method

To provide a detailed exposition of the methodology employed in this paper for detecting inconsistent text–image pairing, this section comprehensively describes the model architecture during the training and testing phases.

### 3.1. Text and Image Preprocessing

Due to the fact that news headlines mostly contain proper nouns, such as locations, the names of individuals, and country names, to enhance the accuracy of the alignment between headlines and image objects, we employ Spacy for Named Entity Recognition (NER) to replace these proper nouns. Simultaneously, we utilize data augmentation techniques on images, including rotation, the addition of noise, translation, photometric transforms, etc. The added noise is in the form of Gaussian noise. Photometric transforms include adjustments in brightness, hue, saturation, and contrast.

### 3.2. Contrastive Learning-Guided Image–Text Matching Training

Inspired by the baseline model on COSMOS [58], we extract features from images and texts separately and interact with them to learn their matching. The training procedure is shown in Figure 2 and described in detail as follows.

**Contrastive Learning Module.** For each image, we use pre-trained Masked-RCNN [59] as the object detection [60] backbone to detect objects included in the image. Then, we feed images and their detected objects (bounding boxes) into the augmentation module.

In the augmentation module, we enhance each identified object by applying various modifications. These augmented images are subsequently passed through an image encoder, which is connected to a fully connected layer to produce a dense vector. For training the contrastive learning model, we designate all augmented images from the same sample as positive instances while randomly selecting images from different samples in the dataset as negative instances. Specifically, applying Mask RCNN to the input image, we can obtain N detected objects to form a set $\{x_k\}_{k=1}^N$ of objects, where $x_k \in \mathbb{R}^{d_x}$, and $d_x$ is the dimension of detected object $x$. Then, we apply data augmentation twice to obtain $2N$ objects, $\{\widetilde{x}_l\}_{l=1}^{2N}$, where $\widetilde{x}_{2k}$ and $\widetilde{x}_{2k-1}$ are two random augmentations of $x_k (k = 1, \cdots, N)$. Different augmentation strategies can be used, such as rotations, the addition of noise, translation, photometric transforms, etc. Thus, an object can be augmented to generate more than two augmented objects, but only two augmented objects are used for each detected object in our setting. We use the following notation for two related augmented objects. Let $i \in I := \{1, ..., 2N\}$ be the index of an arbitrary augmented object, and let $j(i)$ be another augmented object that shares the same source object as the $i$-th augmented object. We feed augmented objects to an object encoder, represented by $E(\cdot) : \mathbb{R}^{d_x} \to \mathbb{R}^d$ ($d$ is the output dimension of $E$), which is a ResNet-50 backbone followed by three components: RoIAlign, average pooling, and two fully connected (FC) layers. Then, we can obtain a 300-dimensional vector for each augmented object, which maps the object feature representation into the application space of the contrastive loss, i.e., $\widetilde{z}_i = E(\widetilde{x}_i)$ and $\widetilde{z}_{j(i)} = E(\widetilde{x}_{j(i)})$ for two augmented objects from the same source object.

To minimize the discrepancy between encoder vectors $\widetilde{z}_i$ and $\widetilde{z}_{j(i)}$ that correspond to the same source object and to maximize the dissimilarity between $\widetilde{z}_i$ and an augmented object from a different source object, we employ self-supervised contrastive learning. This approach allows us to quantify the self-supervised contrastive loss $\mathcal{L}_{CL}$ using the following formulation:

$$\mathcal{L}_{CL} = \frac{-1}{|I|} \sum_{i \in I} \log \frac{\exp(\widetilde{z}_i \cdot \widetilde{z}_{j(i)} / \tau)}{\sum_{a \in A(i)} \exp(\widetilde{z}_i \cdot \widetilde{z}_a / \tau)}, \tag{1}$$

where $|I|$ is the cardinality of $I$, $\tau \in \mathbb{R}^+$ is a positive scalar temperature parameter, $\cdot$ is the inner (dot) product operator, and $A(i) := I \setminus \{i\}$. It is common to regard $i$ as an anchor. $j(i)$ is called the *positive*, and the other $2N - 2$ indices ($\{k \in A(i) \setminus \{j(i)\}\}$) are called the *negatives*. The numerator in the log function of Equation (1) is the representation distance between $\tilde{z}_i$ and $\tilde{z}_{j(i)}$. The denominator is the representation distance between $\tilde{z}_i$ and a total of $2N - 1$ terms, including the positive and negatives. With this contrastive learning module, we can enhance the accuracy of representations from the encoder.

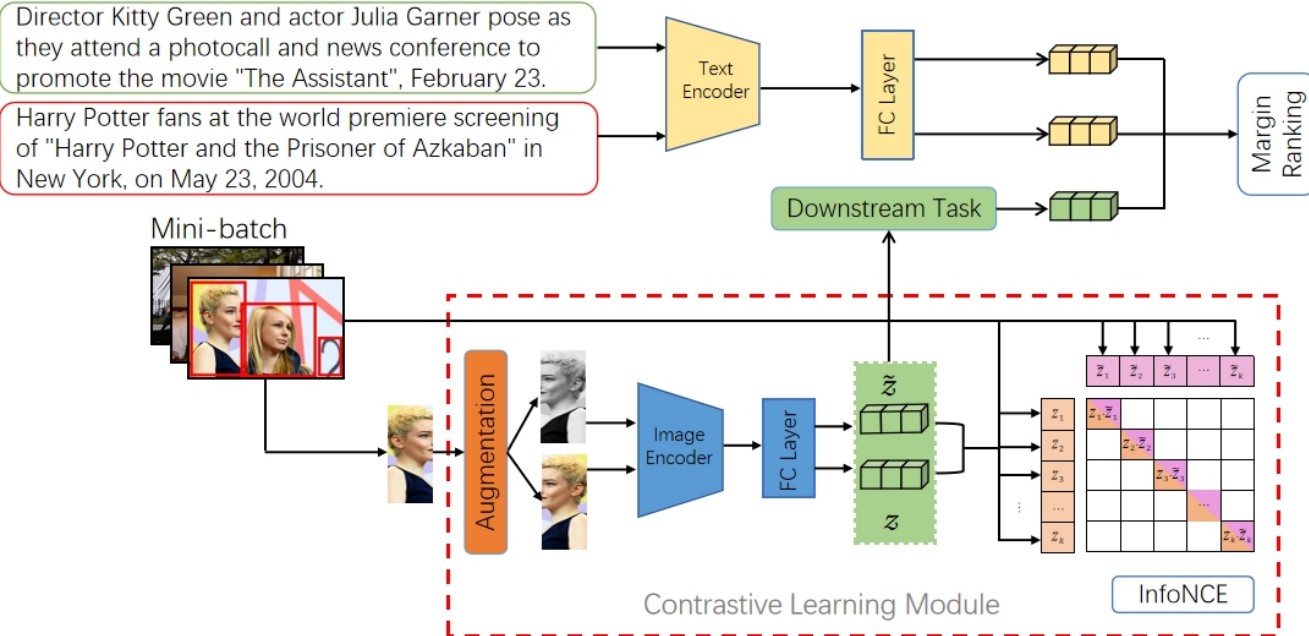

**Figure 2.** The multimodal contrastive learning model consists of two main branches: image feature learning and classification training. In the image feature learning branch, a mini-batch of images (with a size of 64) is inputted into a Mask RCNN, which detects ten objects in each image. Each object is then individually augmented and passed through a fully connected layer to obtain two dense vectors, denoted by $\tilde{z}$ and $z$. Next, a matrix is constructed for the mini-batch, where pairs of $z$ and $\tilde{z}$ from the same object are considered positive instances (located on the diagonal of the matrix). Conversely, all other pairs are treated as negative instances. This matrix is used for training with the InfoNCE loss, as defined in Equation (1). In the classification training branch, two captions are used: a matched caption (shown in green) and another caption randomly sampled (shown in red). These captions are encoded using the pre-trained text encoder. The outputs of the text encoder, combined with the outputs of the image encoder, are used to compute similarities between object–caption pairs. These similarities are then utilized to calculate the margin ranking loss, as described in Equation (1).

**Image–Text Matching Module**. This module aims to match an image with its corresponding text (caption). To achieve this, we employ a pre-trained transformer-based Universal Sentence Encoder (USE, denoted by $U(\cdot)$) [61], as described in [58]. The USE encodes captions into unified 512-dimensional vectors. These vectors are then passed through an additional text encoder ($T(\cdot)$) to convert them into a specific feature space $\mathbb{R}^d$ that matches the output dimension (denoted by $d$) of the image encoder $E(\cdot)$. The text encoder consists of a ReLU activation function followed by a fully connected (FC) layer. As a result, we can represent the final embedded features of the matched caption $c_m$ as $\tilde{c}_m = T(U(c_m))$ and the final embedded features of the random caption $c_r$ as $\tilde{c}_r = T(U(c_r))$.

Then, we evaluate the match performance of the object embedding and the caption embedding. Specifically, we use the dot product to calculate the similarity between $\tilde{z}_i$ and $\tilde{c}_m$ (or $\tilde{c}_r$). We extract the maximum value as the final similarity score:

$$s_m = \max(\{\widetilde{z}_i^\top \widetilde{c}_m | i \in I\}),$$
$$s_r = \max(\{\widetilde{z}_i^\top \widetilde{c}_r | i \in I\}), \tag{2}$$

The final similarity score $s_m$ represents the similarity of the matched caption, while $s_r$ represents the similarity of the random caption. Our objective is to maximize $s_m$ and minimize $s_r$ as much as possible. To achieve this, we devise the following max-margin loss function for training the model:

$$\mathcal{L}_{Match} = [s_r - s_m + \gamma]_+, \tag{3}$$

where $[a]_+ = \max(0, a)$ is the hinge function. $\gamma \in \mathbb{R}$ is a preset margin hyperparameter. The algorithm is shown in Algorithm 1.

---

**Algorithm 1** Out-of-context matching

---

**Data:** sample $\{X_k\}_{k=1}^N$ in batch size $N$
$X_k = < caption_r > image < caption_m >$
$\tau$ and $\gamma$ are constant
**for all** $k \in \{1, \dots, N\}$ **do**
    $O_m \leftarrow \text{MaskRCNN}(X_k)$                                           ▷ 10 object detections
    $C_{km} \leftarrow t \cdot (X_{km})$                                      ▷ text encoding for matched
    $C_{kr} \leftarrow t \cdot (X_{kr})$                                       ▷ text encoding for random
    **for all** $m \in \{1, \dots, 10\}$ **do**
        $\{A, \tilde{A}\} \leftarrow A(O_m)$                                   ▷ augmentation
        $Z \leftarrow f \cdot (A)$                                        ▷ image encoding
        $\tilde{Z} \leftarrow f \cdot (\tilde{A})$
    **end for**
**end for**
$M = N * 10$                                           ▷ # of augmentation in batch
**for all** $i \in \{1, \dots, 2M\}$ and $j \in \{1, \dots, 2M\}$ **do**
    $s_{i,j} = z_i z_j / (\|z_i\| \|z_j\|)$                                  ▷ pairwise similarity
**end for**
**define** $\ell(i, j)$ **as** $\ell(i, j) = -\log \frac{exp(s_{i,j}/\tau)}{\sum_{k=1}^{2M} 1_{[k \neq i]} exp(s_{i,k}/\tau)}$
$\mathcal{L}_{CL} = \frac{1}{2M} \sum_{k=1}^M [\ell(2k-1, 2k) + \ell(2k, 2k-1)]$
**define** $s_m = \max(Z_k \cdot C_{km} | k \in \{1, \dots, N\})$
**define** $s_r = \max(Z_k \cdot C_{kr} | k \in \{1, \dots, N\})$
$\mathcal{L}_{Match} = [s_r - s_m + \gamma]_+$
update networks $f$ and $t$ to minimize $\mathcal{L}_{CL}$ and $\mathcal{L}_{Match}$

---

**Cross-Training**. We first train object encoder $E$ in the contrastive learning module based on $\mathcal{L}_{CL}$ (Equation (1)) for all images in the dataset. Then, we fix the contrastive learning module and train text encoder $T$ according to $\mathcal{L}_{Match}$ (Equation (3)) on all images. The weights of the whole model are updated iteratively.
**Joint Training**. In addition to cross-training, we explore joint training as well. Rather than freezing one of the loss functions during training, we normalize the loss of contrastive learning module $\mathcal{L}_{CL}$ (Equation (1)) and add $\mathcal{L}_{Match}$ (Equation (3)) to it to obtain the overall average loss on all images.

### 3.3. Image–Text Mismatching Prediction

Given testing data (an image and two captions, e.g., *image *), to determine whether there is a mismatch between the image and text in our model, we adopt the approach outlined in [58], as depicted in Figure 3. This prediction relies on two scores: the Intersection over Union (*IoU*) score and the Sentence BERT (SBERT) score ($S_{sim}$). The *IoU* is the overlap ratio of the most closely matching objects between *caption_1* and *caption_2* in the image, representing the similarity between the objects corresponding to

the two captions. A higher similarity indicates a higher likelihood that the two captions describe the same object in the image. Meanwhile, the SBERT score measures their similarity in terms of sentence semantics.

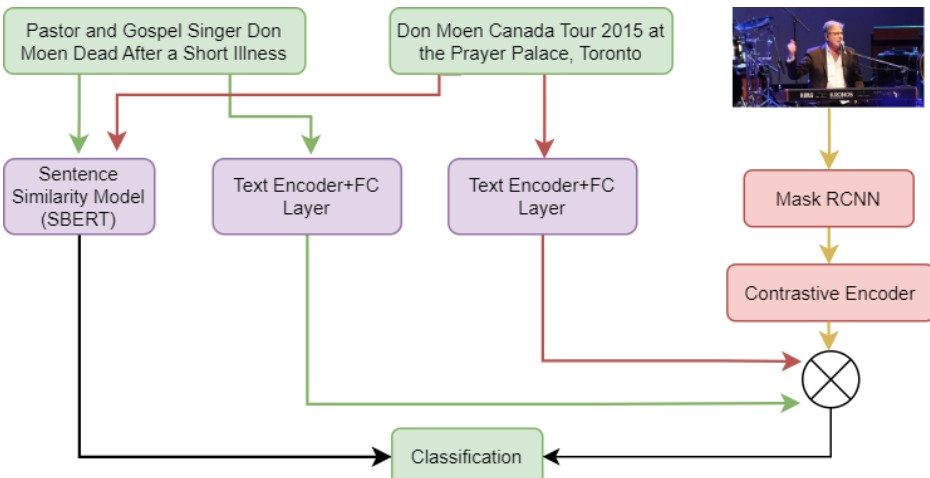

**Figure 3.** The testing structure. *IoU* indicates whether the two captions are describing the same object, and $S_{sim}$ represents the semantic similarity between the two captions. It predicts out-of-context content if both scores are higher than their preset thresholds.

Specifically, we utilize the state-of-the-art SBERT model [62] to assess the similarity between two captions. This model is pre-trained on the Sentence Textual Similarity (STS) task and enables us to obtain the ($S_{sim}$) score, which assesses both the semantic and syntactic similarities between the two sentences. The SBERT model takes two text inputs and produces a score ranging from 0 to 1. A higher score indicates a greater degree of shared contextual similarity between the two captions.

To compute the *IoU* score, we utilize both the image encoder and the text encoder derived from the trained language–vision model. Firstly, we determine the visual correspondences of objects $B_{IC_i}$ in the image for each caption. For instance, $B_{IC_1}$ represents the highest alignment value (object) between image $I$ and *caption_1*. Subsequently, we calculate the *IoU* score in the range [0,1] by evaluating the overlap between the bounding boxes (areas) corresponding to *caption_1* and *caption_2*. Finally, during the assessment, we set the threshold to 0.5 and consider both the *IoU* and $S_{sim}$ scores. Only when the *IoU* is greater than the threshold and $S_{sim}$ is less than the threshold can we determine it to be out of context (OOC); otherwise, it is not out of context (NOOC). We employ the following rule to predict whether an image–caption pair is out of context:

- Out of context if $IoU(B_{IC_1}, B_{IC_2}) > threshold$ and $S_{sim} < threshold$;
- Not out of context otherwise.

## 4. Experimental Evaluation

### 4.1. Datasets and Preprocessing

As previously mentioned, the lack of a gold-standard training dataset presents a significant challenge in our research endeavors. We selected a publicly available, extensive dataset to solve this problem. Aneja et al. [28] compiled this dataset by aggregating information from two primary sources: fact-checking websites and various prominent mainstream news platforms, like the *New York Times*, CNN, Reuters, and others. The original dataset is stored in JSON format, and its structure follows a pattern where each data entry comprises an image accompanied by two captions: one representing a genuine description and the other a synthetic one. A summary of the dataset's key characteristics is presented in Table 1.

**Table 1.** The statistical summary of the datasets.

|                 | # of Images | # of Captions | Annotation |
| --------------- | ----------- | ------------- | ---------- |
| Training Data   | 160 k       | 360 k         | no         |
| Validation Data | 40 k        | 90 k          | no         |
| Testing Data    | 1700        | 1700          | yes        |

While we recognize that the training and validation sets lack labels, as the synthetic captions for each image were randomly selected from other samples, the testing set differs. In the testing set, the text–image pairs have been meticulously labeled by the authors. We depict an example of a data instance in Figure 1.

In our text preprocessing, we implemented entity extraction to substitute all names, locations, and dates with unique tokens. For instance, consider the caption, "An image gives a genuine glimpse at Florida congressman Matt Gaetz' hair". After our preprocessing, it becomes: "An image gives a genuine glimpse at GPE congressman PERSON' hair".

*4.2. Experimental Setup*

In this section, we provide a detailed overview of our experimental setup. Our study encompasses three main experiments aimed at showcasing the advantages of contrastive learning, especially when dealing with limited training data, as well as introducing a novel model for distinguishing the veracity of captions. Compared to our preliminary version [29], we explore the impact of different text encoding layers on model performance. To be more precise, our research involves a comparison of three distinct models:

- *Baseline*: The model originates from [58].
- *Cross-training*: There are two loss functions—*InfoNCE* for contrastive learning and *MarginRanking* for classification. We independently optimize each of the two loss functions to find the optimal match.
- *Joint training*: As the comparative experiment with alternating training, we simultaneously optimize both loss functions, *InfoNCE* and *MarginRanking*, with designed weights.
- *Different encoding*: To investigate the impact of different pre-trained language models on our experiments, we replaced the text encoding part with BERT [63] and Vicuna [64] for comparison.

  **Evaluation Metrics.** Given that the ultimate goal of this paper is to boost the ability to detect OOC content, we use the standard classification evaluation metrics: (accuracy, precision, recall, and F1-score).

  **Implementation Details.** Furthermore, we understand that detecting out-of-context (OOC) content involves making a trade-off decision. We must weigh the potential consequences of false negatives, where OOC content is mistakenly classified as non-OOC, against the false positives, where non-OOC content is incorrectly identified as OOC. In a real-world scenario, it is presumed that the failure to identify misinformation and allow its propagation would have a more significant impact on social networks than mistakenly labeling clean content as misinformation. Consequently, our research will expand upon the original study, which solely focuses on accuracy, by placing greater emphasis on recall, also known as the true positive rate.

The language model BERT utilizes pre-trained weights from bert-base-uncased, while the large language model Vicuna uses pre-trained weights from Vicuna V0 7B. To account for the random initialization of the neural network, we conducted a total of three experiments and calculated the averages of the results. Furthermore, we applied default hyperparameters in our model, which included utilizing the Adam optimizer, employing the ReLU activation function, setting a batch size of 64, and training the model for 10 epochs.

### 4.3. Contrastive Learning vs. Baseline

We would like to investigate whether contrastive learning is able to achieve better performance on a small-sized training dataset compared to the baseline model. The specific configurations we employed were as follows:

- We detected ten objects using the Mask-RCNN model.
- To introduce variation in the training data, we incorporated augmentation techniques such as rotation, the addition of gray, filtering, resizing, translation, brightness adjustment, and more.
- The ResNet model consists of 18 convolutional layers, with the output being a 512-dimensional feature representation (the text encoder also has a dimension of 512).
- Lastly, the Dense Layer produced a dense vector output of 300 dimensions.

Table 2 presents the results, highlighting the top score for each evaluation metric. It is evident that the baseline model consistently performs the worst across all four metrics. As expected, replacing the convolutional network's structure with a contrastive learning module in the image encoder leads to improved out-of-context (OOC) content detection. Both the cross-training and joint training models have an accuracy of more than 80, indicating an approximately 10% enhancement compared to the baseline model's accuracy of 73.53%. Regarding recall, the contrastive learning models outperform the baseline's 87.76%, with the cross-training and joint training models achieving recall scores of 90.94% and 92.23%, respectively. This improvement can be attributed to the enlarged training samples achieved through augmentation. These results suggest that contrastive learning models effectively address the challenge of insufficient training data. However, according to the paired *t*-test, the difference between the performance of cross-training and joint training contrastive models is not statistically significant (*p*-value is 0.718).

In order to validate the impact of text encoding on model performance, we added BERT and Vicuna, using joint training as a comparison. Based on the performance across four metrics, the difference in performance after replacement was negligible. This indicates that solely replacing the text encoding part does not have a decisive effect on this model.

**Table 2.** The comparison results of five models for accuracy, precision, recall, and F1-score.

|  | **Accuracy** | **Precision** | **Recall** | **F1-Score** |
|---|---|---|---|---|
| Baseline | 73.53 | **75.8** | 87.76 | 80.56 |
| Cross-Training | 80.23 | 75.3 | 90.94 | 82.05 |
| Joint Training | **80.47** | 74.85 | **92.23** | **82.45** |
| BERT | 80.4 | 74.35 | 91.76 | 82.14 |
| Vicuna | 80.28 | 74.54 | 92 | 82.35 |

We acknowledge that the accuracy of the methods mentioned above is not as high as the results presented in the original paper [58]. This is primarily due to our random selection of a smaller subset of the training data, which comprised fewer than 5000 samples. Compared to the reported accuracy of 85% achieved by utilizing the complete training set of 160,000 samples, our findings demonstrate that contrastive learning can achieve approximately 94% (0.80/0.85) of the performance using only 3% (4.5 k/160 k) of the training data. Additionally, it is worth noting that the marginal improvements obtained by incorporating more data for training would be limited.

### 4.4. Comparison on Varying Training Data Sizes

Building upon the results obtained in the previous experiment (Section 4.3), we aim to further investigate the impact of different training sizes on the performance of our models. Specifically, we are interested in determining whether contrastive learning can still achieve comparable results when trained on even smaller amounts of data, and how varying amounts of training data affect the classification ability of the models. To accomplish this, we created 10 different levels of training sizes, ranging from 500 to 5000 samples in

intervals of 500. These samples were randomly selected from the original dataset. For each training size level, we used the same training data across all five models. Finally, we measured the classification accuracy on the testing set using the same methodology as in the previous experiment. The results of these evaluations are depicted in Figure 4. This analysis allows us to assess the impact of the training data size on the classification performance of our models.

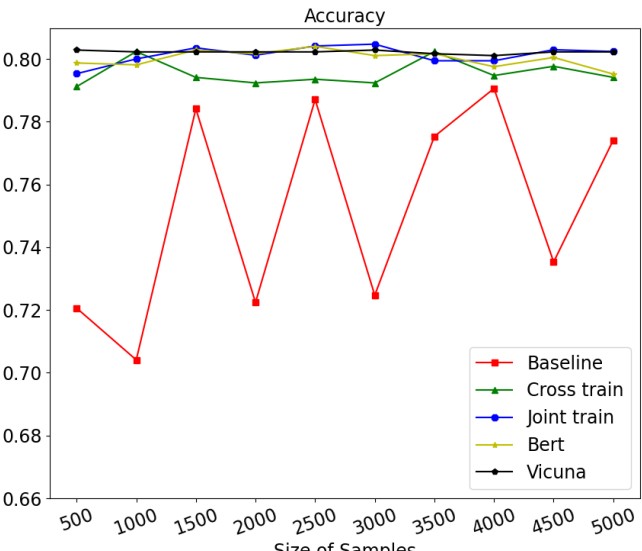

**Figure 4.** OOC classification accuracy for five different models with varying sizes of training samples.

In theory, increasing the amount of training data should lead to improved classification performance for models. However, the results depicted in Figure 4 reveal an unexpected trend for the baseline model. Contrary to expectations, the baseline model's accuracy does not consistently increase as the training data size grows. Instead, it shows fluctuations and even decreases in performance at certain training data sizes. For example, the highest accuracy of the red line is achieved at 1500 training samples, reaching 0.78. Surprisingly, the baseline model experiences significant drops in accuracy when trained on over 2000, 3000, and 4500 samples, with the accuracy decreasing from nearly 0.78 to 0.72. Despite these fluctuations, the baseline model exhibits an overall upward trend, with the red line increasing from 0.72 to 0.77. These findings highlight the complex relationship between the training data size and classification performance, suggesting that adding more data may not always lead to consistent improvements and could even result in unexpected performance fluctuations.

In contrast, the two contrastive learning models demonstrated remarkable performance (around 0.79) even when trained on a limited number of data samples. Notably, both models consistently achieved accuracies ranging from 0.78 to 0.80, surpassing the lowest accuracy of 0.72 obtained by the baseline model. Furthermore, the performance of the contrastive learning models exhibited a steady improvement as more data were added during training, albeit with limited gains. These findings highlight the effectiveness of contrastive learning in leveraging small training datasets and their ability to achieve superior classification results compared to the baseline model.

In general, both contrastive learning models exhibited similar performance, making it difficult to determine which one was superior. We also note that replacing the SOTA text encoder such as BERT or Vicuna would improve the accuracy, particularly with the smallest training size (500).

To gain further insights, we also evaluated the precision, recall, and F1-score, as shown in Figure 5. The joint training approach yielded the highest recall and F1-score, indicating its superior ability to correctly identify true positives and achieve a balance between precision and recall. The cross-training approach followed closely, demonstrating competi-

tive performance in terms of recall and F1-score. On the other hand, the baseline model exhibited the lowest performance across all metrics, indicating its limitations in accurately classifying the correct captions. These findings emphasize the advantages of the contrastive learning models over the baseline model, particularly in terms of recall and F1-score.

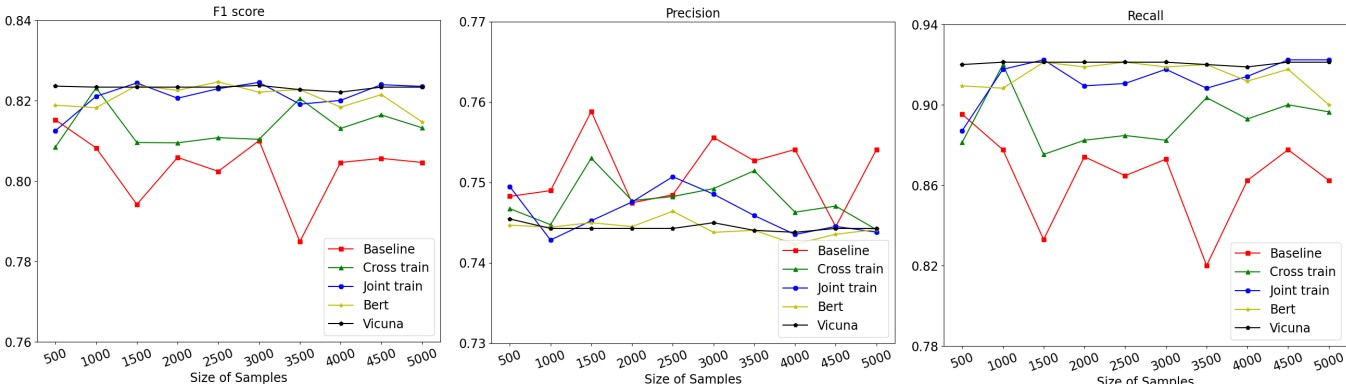

**Figure 5.** OOC classification F1-score, precision, and recall for three different models with varying sizes of training samples.

The performance metrics of different text encoding models exhibit stable curves, as shown in the yellow and black lines in the figure. We assume that this is because complex encoders are better able to extract features intrinsic to the text.

### 4.5. Contrastive Learning for True Caption Classification

We have observed that the sentence similarity BERT model, which was pre-trained without fine-tuning, plays a dominant role in the classification process. Figure 3 illustrates that the final prediction depends on both the *IoU* and $S_{sim}$ scores. However, a significant majority of the testing data (over 80%) exhibit an *IoU* value of approximately 0.9, surpassing the threshold of 0.5. As a result, the final classification is primarily influenced by the $S_{sim}$ score. In order to address this bias, we propose the direct utilization of contrastive learning during the training phase. This approach aims to alleviate the overreliance on the sentence similarity BERT model and improve the overall classification performance.

We have made a simple modification to our model for the task of caption classification, as depicted in Figure 6. The process begins by extracting the vector representation of the image from the trained contrastive learning model. Next, we calculate the cosine similarity between this representation and the vector representations of two captions: one that matches the image and another that does not. Based on the similarity score, we determine the true caption, with the matched caption expected to have a higher similarity value with the image. In our former experiment, we identified contrastive learning + Vicuna + joint as the best combination, so we used that for classification here. However, the results obtained using this method are mediocre, with only 941 out of 1700 (55%) captions correctly detected. We acknowledge the room for improvement and will continue to work on enhancing this approach in our future work.

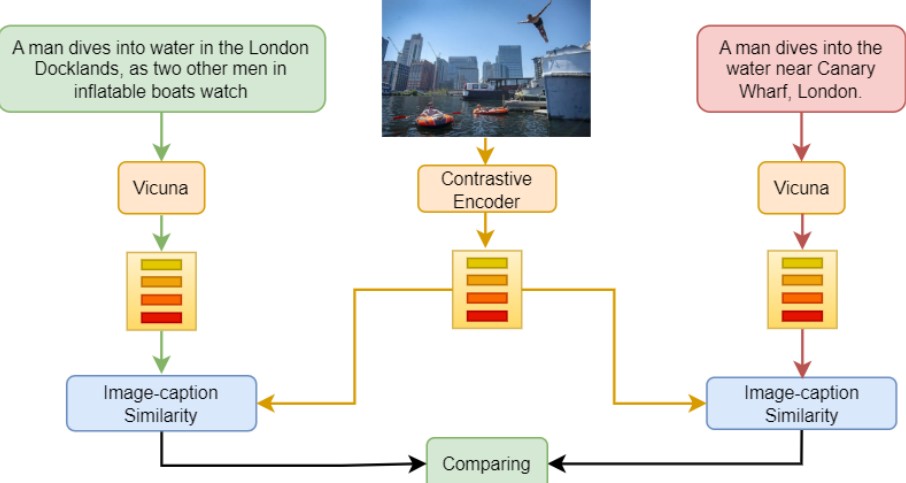

**Figure 6.** To classify the true caption based on the image–caption model, we utilize the pre-trained contrastive learning module (refer to Figure 2) for extracting image features. Additionally, we encode two captions separately using Vicuna. The resulting caption vectors are then compared to the image vector to compute their similarity. The prediction is based on the similarity score, where the image vector is closer to the true caption than the false one. This approach allows us to determine the correct caption for a given image by leveraging the similarity between the image features and the encoded captions.

## 5. Conclusions

This work primarily focused on examining the performance of contrastive learning in addressing the challenge of limited labeled data in the context of text–image pairing. The key points highlighted in this study include the following:

- The proposal of an advanced model: We introduced an improved out-of-context (OOC) detection model by leveraging contrastive learning, a self-supervised machine learning technique. Our model incorporates data augmentation during training, resulting in superior performance compared to the benchmark model outlined in the original paper.
- Emphasis on inadequate labeled data: We specifically investigated the scenario where there is a lack of labeled data, a common limitation in many classification tasks. Through our comparisons, we demonstrated that contrastive learning exhibits strong capabilities in learning image features, achieving 94% of the full performance, even with a significant reduction in the training data size.
- A comprehensive analysis of classifiers: We conducted a thorough analysis of different classifiers' abilities to handle varying training data sizes. The results showcased the stability and consistent performance improvement of the contrastive learning model as more training data were added. In contrast, the baseline model exhibited fluctuating results with ups and downs.
- A comparison of the impact of different text encoders on model performance: The proposed method expands on our previous work [29] in terms of textual content and replaces the text encoding module with the text encoder of a large-scale model, showing better stability.

In summary, our study highlights the effectiveness of contrastive learning in addressing the challenge of limited labeled data, providing stable performance and consistent accuracy improvements when training data are augmented.

Lastly, it is important to acknowledge a limitation of the COSMOS dataset, which relates to the labeling of out-of-context (OOC) instances. The OOC label is determined based on whether the two captions align with each other and correspond to the given image. This approach overlooks the real-world scenario of accurately identifying which caption is true. Although we proposed a new classifier model to address this issue, the results

obtained in our experiments were not promising. This highlights the complexity and challenges associated with accurately determining the actual caption in the context of OOC detection. Further research and improvements are needed to enhance the performance of models in this regard.

In future work, we will continue to explore the ability of contrastive learning to extract multimodal features such as text and images. Additionally, we intend to enhance the accuracy of misinformation detection by refining the fusion of multimodal information.

**Author Contributions:** Conceptualization, H.C. and P.Z.; methodology, X.W. (Xin Wang); software, P.Z.; validation, P.Z., X.W. (Xin Wang) and S.H.; formal analysis, H.C.; investigation, S.H.; resources, X.W. (Xin Wang); data curation, B.Z. and S.L.; writing—original draft preparation, P.Z. and H.C.; writing—review and editing, H.C. and C.-S.L.; visualization, G.H.; supervision, X.W. (Xi Wu); project administration, H.C.; funding acquisition, J.H. All authors have read and agreed to the published version of the manuscript.

**Funding:** This research was funded by the Sichuan Science and Technology Program (No. 2022YFQ0073, No. 2023YFQ0072), and the Key Lab of Internet Natural Language Processing of Sichuan Provincial Education Department (No. INLP202203).

**Data Availability Statement:** The data presented in this study are available in COSMOS [28].

**Conflicts of Interest:** The authors declare no conflicts of interest.

## Abbreviations

The following abbreviations are used in this manuscript:

COSMOS    Catching Out-of-Context Misinformation

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
