# Peer review of "Few-Shot Learning for Misinformation Detection Based on Contrastive Models"

_electronics, doi:10.3390/electronics13040799_

Round 1
Reviewer 1 Report
Comments and Suggestions for Authors
Manuscript ID: electronics-2814247
Journal: Electronics
Title: Few-shot Learning for Misinformation Detection based on Contrastive Models
Authors: Peng Zheng , Hao Chen , Shu Hu , Bin Zhu , Jinrong Hu * , Ching-Sheng Lin , Xi Wu , Siwei Lyu , Guo Huang , Xin Wang
Fake news has become more prevalent as social media has grown, and this has had a huge impact on people's lives as well as society. It is challenging to report news objectively under the censors' constraints. Most studies employ supervised techniques, which reduce the efficacy of the identification of fake news by depending on a substantial volume of labelled data. Although the real fake news is more often presented as text-image pairs, the focus of these experiments is on the identification of fake news in a single modality, such as text or picture. Authors presented a contrastive learning-based self-supervised model in this research. Using dot product graphic matching, this approach allows for simultaneous feature extraction for both text and images. It improves image feature extraction using contrastive learning, resulting in a strong visual feature extraction capacity with less training data needed. The COSMOS false news dataset is used to compare the model's efficacy to the baseline. The results show that just about 3 percent of the data is used for training for detecting bogus news with mismatched text-image pairs. With the entire amount of data used for training, the model attains an accuracy of 80%, which is equal to 95% of the original model's performance. Specifically on the COSMOS dataset, altering the text encoding layer improves experimental stability and offers a significant improvement over the original model.
The paper is well written, interested and the results are good, I would like to suggest the following MINOR corrections before acceptance:
ــــــــــــــــــــــــــــــــــــــــــــــــــــــــــــــــــــــــــــــــــــــــــــــــــــــــــــــــــــــــــــــــــــــــــــــــــــــــــــــــــــــــــــــــــــــــــ
(1) A professional proofreading revision is strongly required. Typos must be corrected.
(2) Please add more details about the studied model
(3) The introduction must be reformulated to contain literature and future works, the main aim of the work.
(4) The arrangement of the manuscript should be added in a paragraph at the end of the introduction.
(5) The authors should state clearly in the introduction the advantages of the used technique and a summary of the literature.
(6) I didn’t see any mathematical model in the paper, if there is this will improve the paper, as some works regarding the mathematical modelling can be added, for example (or others no problem):
https://doi.org/10.3934/math.20231592
https://doi.org/10.1016/j.aej.2023.10.003
https://doi.org/10.3934/math.20231569
https://doi.org/10.3934/math.20231592
(7) The authors should revise and carefully arrange the references according to the guidelines of the journal.
(8) The arrangement of the manuscript should be added in a paragraph at the end of the introduction.
After these considerations, I think the paper can be accepted.
Comments on the Quality of English Language
There is no comments
Reviewer 2 Report
Comments and Suggestions for Authors
Advantages.
Good practical research in the field of the detection of fake news. The topic of this article is interesting and meaningful for misinformation detection. The article is a continuation of the previously published work.
The design of the manuscript is well structured:
- Introduction part is given.
- The methodology part with algorithms is given (Related works).
- Experimental results and analysis part is given.
- Conclusion part is given.
There are no significant criticisms about the research methodology.
Disadvantages:
The article is more about pattern recognition than topics of Electronics.
Some comments:
- Position figures as soon as possible after they are first cited in the text (good style for scientific manuscripts). In line 364 you cited Fig.5, but in line 392 – Fig.4.
- The caption of figures 2 and 6 is so long.
- References should be numbered in order of appearance.
- No literature references found in the text: 5,6,7,9,18,20,22,23,40,41,44,45,47,54,57.
- Line 17. Mistake “depp learning”.
Additional comments:
1. The main question of these studies is on the detection of fake news in the form of text-image pairs.
2. This research is original – the authors introduced a self-supervised model grounded in contrastive learning which gives possibilities for examining the performance of contrastive learning in addressing the challenge of limited labeled data in the context of text-image pairing.
3. There are no significant criticisms about the research methodology. Good experimental results are given. The authors demonstrated that contrastive learning exhibits strong capabilities in learning image features and achieving 94% of the full performance.
4. The conclusions are general, and it is necessary to specify more detailed future research directions.
5. References should be numbered in order of appearance and some references not be found in the text.
Reviewer 3 Report
Comments and Suggestions for Authors
I have read this paper and have some editorial corrections that need to be made to the References listed that currently are inconsistent in the formats used. I have read the body of the paper and have no comments or corrections to be made. I cannot comment on the correctness of Algorithm 1 on page 6.
Format of References on pages 13 to 16: Some of the References used capital letters for each of the major words of Titles, and some other References do not and instead only use capital letter for the first word of the Title of article and/or the title of the Conference Proceedings or journal name.
For example:
(1.) Reference 1 and 33 and 47 use capital letter for each of the major words of both Title of article and also name of Conference Proceedings when Reference 7and 22 and 23 and 25 and 40 do not use capital letter for any word except the first word in title of paper and title of Conference Proceedings. Reference 22 also uses word "Proceedings" twice in the title and should only be once. There are many others such as References 35 and 46 and so a very careful editorial review needs to be made for each of the 61 References provided with also noting another inconsistent format as described in item (2.) below.
(2.) References 12 and 16 and 18 and 24 and 59 and 61 and possibly others use capital letters for each of the major words of the title of Conference Proceedings but do NOT use this format for the titles of article and use capital for first word only of the title of article. Once again, a very careful editorial review needs to be made for each and every of the 61 References list to be certain the correct format for this journal is used. The authors need to find the "Directions for Authors for submission" for the Electronics journal or all MDPI publications for this paper to be acceptable for publication.
Reviewer 4 Report
Comments and Suggestions for Authors
I come from a different background/perspective in AI than yours so my comment may not be as valid. However, I am not really convinced that the methodology in "ACM Multimedia Grand Challenge on Detecting Cheapfakes" is sound (they are comparing a real caption with a random caption as opposed to a caption that is fake and fabricated to look real). Maybe an explanation of why these approaches are effective in real life could convince newcomers like me.
Comments on the Quality of English LanguageA few writing errors like "we aims to focusing towards".
They could have easily been picked up using linters or grammar checkers.
Reviewer 5 Report
Comments and Suggestions for Authors
To improve this paper:
1. Please, sort the References in the right order 1,2,3,..etc (the article begins with link No. 50, page 1)
2. Please check the data of Fig. 4 vs Table. 2 (Precision of Baseline/Crosstraining)
3. The References contains too many links to ArXive.org; it would be better to replace some of them by published articles.
4. Please improve chapter 3.2 - more detailed description is needed.
Comments on the Quality of English LanguageMinor editing of English language required
Round 2
Reviewer 2 Report
Comments and Suggestions for Authors
Now all is OK